# Analyses of Transcriptomics Cell Signalling for Pre-Screening Applications in the Integrated Approach for Testing and Assessment of Non-Genotoxic Carcinogens

**DOI:** 10.3390/ijms232112718

**Published:** 2022-10-22

**Authors:** Yusuke Oku, Federica Madia, Pierre Lau, Martin Paparella, Timothy McGovern, Mirjam Luijten, Miriam N. Jacobs

**Affiliations:** 1The Organisation for Economic Cooperation and Development (OECD), 2 Rue Andre Pascal, 75016 Paris, France; 2European Commission, Joint Research Centre (JRC), Via Enrico Fermi, 2749, 21027 Ispra, Italy; 3Istituto Italiano di Tecnologia, 16163 Genova, Italy; 4Institute of Medical Biochemistry, Biocenter, Medical University of Innsbruck, Innrain 80, 6020 Innbruck, Austria; 5US Food and Drug Administration (FDA), 10903 New Hampshire Avenue, Silver Spring, MD 20901, USA; 6Centre for Health Protection, National Institute for Public Health and the Environment (RIVM), Antonie van Leeuwenhoeklaan 9, Bilthoven, 3721 MA Utrecht, The Netherlands; 7Centre for Radiation, Chemical and Environmental Hazard (CRCE), Public Health England (PHE), Chilton OX11 0RQ, Oxfordshire, UK

**Keywords:** integrated approaches to testing and assessment (IATA), non-genotoxic carcinogen(s) (NGTxC(s)), gene expression, transcriptomics, carcinogenicity, organisation for economic co-operation and development (OECD)

## Abstract

With recent rapid advancement of methodological tools, mechanistic understanding of biological processes leading to carcinogenesis is expanding. New approach methodologies such as transcriptomics can inform on non-genotoxic mechanisms of chemical carcinogens and can be developed for regulatory applications. The Organisation for the Economic Cooperation and Development (OECD) expert group developing an Integrated Approach to the Testing and Assessment (IATA) of Non-Genotoxic Carcinogens (NGTxC) is reviewing the possible assays to be integrated therein. In this context, we review the application of transcriptomics approaches suitable for pre-screening gene expression changes associated with phenotypic alterations that underlie the carcinogenic processes for subsequent prioritisation of downstream test methods appropriate to specific key events of non-genotoxic carcinogenesis. Using case studies, we evaluate the potential of gene expression analyses especially in relation to breast cancer, to identify the most relevant approaches that could be utilised as (pre-) screening tools, for example Gene Set Enrichment Analysis (GSEA). We also consider how to address the challenges to integrate gene panels and transcriptomic assays into the IATA, highlighting the pivotal omics markers identified for assay measurement in the IATA key events of inflammation, immune response, mitogenic signalling and cell injury.

## 1. Introduction

The identification of classical tumour suppressors and oncogenes has given greater depth to the understanding of mechanisms of carcinogenesis, although the underlying mechanisms of carcinogenesis in the majority of cancer types remained unclear until the era of large-scale genomics [1]. Indeed, the emergence of technologies such as microarrays and next generation sequencing (NGS) over the last two decades has enabled the systematic analyses of the genomic and transcriptomic landscapes of various cancer types, and the identification of key gene alterations, including mutations, fusions, epigenetic silencing, copy number alterations, and consequent gene expression changes [2]. These are exemplified by a number of analyses from the Cancer Genome Atlas (TCGA) programme and have expanded the understanding of the natural history and the molecular characteristics of cancer progression [3,4,5,6,7,8,9,10]. Key gene alterations can lead to aberrant cellular signalling and biochemical processes, via changes in gene expression that, if persistently expressed, may eventually lead to malignant phenotypes. 

Genotoxic chemicals cause DNA damage or chromosome instability by direct action upon DNA and/or the mitotic apparatus, leading to mutations, i.e., substitutions, frameshifts, small indels or gross chromosome rearrangements, that drive genomic alterations and eventually the carcinogenesis processes [11]. On the other hand, non-genotoxic carcinogens (NGTxCs) have been described for their potential to induce cancers without interacting directly with either DNA or the cellular apparatus involved in the preservation of the integrity of the genome [12,13]. NGTxCs are considered to induce either inflammation, immune suppression, oxidative stress, epigenetic silencing or other changes in biological processes, leading to aberrant cellular signalling and genomic instability which are predominant to their carcinogenic potential. 

The carcinogenicity safety assessment of any substance commonly starts with the assessment of genotoxicity, via an in vitro testing battery (usually a bacterial reverse mutation assay plus mouse lymphoma thymidine kinase (tk) mutation assay (MLA) or hypoxanthine-guanine phosphoribosyltransferase (hprt) test and an in vitro micronucleus test), followed by appropriate in vivo studies. In the case of positive results for chemicals, a long-term carcinogenicity study in rodents may be required depending on the product sector and/or regulatory jurisdiction [14,15], whilst for pharmaceuticals rodent cancer bioassays are often conducted, regardless of genotoxicity results. For industrial chemicals, however, the carcinogenic potential of NGTxCs that may yield negative results in the initial genotoxicity testing battery may go undetected [16,17]. 

The identification of NGTxCs is still a regulatory challenge, one that an OECD expert group has been established to address [12,18]. The expert group has achieved consensus on the structure of the IATA [12] that has its foundations on the common hallmarks of cancers [18,19] with consideration of the key characteristics of carcinogens [20]. A number of biological processes need to be examined within the IATA, and existing information and data resources should be first employed before embarking upon designing testing strategy and testing chemicals. Tools that enable the initial screening of such changes can help with prioritisation of subsequent more complex assays within the IATA decision-making process and tease out the priority markers that will need to be identified, or not, in the more complex assays. In this context, here we review the application of transcriptomics approaches suitable for pre-screening gene expression changes that are associated with phenotypic alterations that underlie the carcinogenic processes. 

We explore how transcriptomics can be applied in the IATA, for regulatory purposes, to identify the changes in the gene expression of particular biological processes or modes of action. Then we can effectively screen for potential non-genotoxic mechanisms of carcinogens at an early stage of the IATA, and thereby target later testing effectively to increase regulatory confidence in the application of these tools. With particular reference to gene expression and cellular signalling, we examine the technical and regulatory challenges to apply transcriptomics tools within the NGTxC IATA. 

## 2. Transcriptomics to Predict Mechanisms of Action of Non-Genotoxic Carcinogens

The advance of technologies has given rise to multiple ‘omics’ themes. Toxicogenomics examines the toxic effects of chemicals in experimental models and humans, and their possible mechanisms are examined by the collective analyses of biomolecules [21]. Transcriptomic analyses provide a tool that is especially powerful for identifying the possible mechanisms of chemicals with unknown characteristics (reviewed in [22]), although caution is necessary due to a great deal of inconsistencies across data interpretation, and improvement in data reproducibility is needed [23,24]. Several studies have successfully applied transcriptomic analyses to identify non-genotoxic mechanisms of chemical carcinogenesis. 

Dean et al., for example, has employed microarray analysis in combination with Gene Set Enrichment Analysis (GSEA) for the evaluation of dose- and time-dependent changes in gene expression after 2,3,4,6-tetrachlorophenol, bromobenzene and N-nitrosodiphenylamine exposure in the rat [25]. GSEA examines the similarity of differential gene expression in a rank with predefined gene sets [26]. In their study, Dean et al. observed enrichment of gene expression patterns of several hallmark gene sets, including those of proliferation, growth signalling and immune responses, upon exposure to any of the three chemicals across a range of doses. Although chemicals can induce gene enrichment in common pathways, the pathways are affected differently depending upon experimental conditions in relation to dose/concentration and exposure duration, as reported in multiple published studies [25,27,28,29]. Importantly, benchmark dose (BMD) modelled expression of leading-edge genes in the most sensitive and commonly enriched pathways has been shown to correlate well with BMD values derived for apical endpoint [25]. It is accepted that utilisation of GSEA can improve accuracy and confidence in the predictions of possible dose- and time-dependent mechanisms of carcinogenesis. 

Transcriptomics analyses are useful for possible phenotypic changes that can occur in the carcinogenic process. Mascolo et al. reported a study where the in vitro cell transformation assay (CTA) was combined with transcriptomic analysis by microarray, and the mechanisms of transformation by the genotoxic chemical 3-methylcholanthrene at different time points and concentrations using mouse BALB/c 3T3 cells were described [30]. In essence, this study demonstrates the added value of combining the CTA with gene expression analysis, for obtaining insight into likely non-genotoxic carcinogenic mechanisms of a chemical of interest, in early and late stages of cell transformation that occur in addition to the known chemical’s genotoxic properties. This approach has also been successfully applied both for mouse BALB/c 3T3 cells [31] and for the Bhas CTA [32]. 

The examples given here show that the effects of NGTxCs can vary with respect to their concentrations/doses and duration of exposure and indicate the differential early/late responses at the gene expression level, and switches in mechanisms. Aspects that need to be expanded upon for appropriate regulatory study design, illustrated in this paper include consideration of the adequacy of:the gene panels and assays (Section 3)the cells or tissues; (Section 4.1)the time points and concentrations or doses (Section 4.2)the derivation of points of departure for human risk assessment purposes (Section 4.3)

Ultimately, testing pathways related to molecular initiating events (MIEs) or key events (KEs) relevant for carcinogenesis should be followed up and anchored to phenotypic assay results for mutual strengthening of the weight of evidence (Figure 1).

## 3. Transcriptomic Assays and Gene Panels to Identify Key Cell Signalling Pathways and Predictive Markers

We first selected commercially or publicly available transcriptomics-based multiplex assays and gene panel lists, with the aim of assessment for potential inclusion in a pre-screening approach relevant for the purpose of the NGTxC IATA. Gene panel lists vary, across different assay types, in composition (e.g., type of transcripts, specific targets) and number of genes. In addition, the specific procedures applied for gene selection, including the intended applications, may define the initial gene candidate lists and final structure of gene panels. A comparison and selection of the appropriate assay types to be integrated in the IATA requires an evaluation of the different study designs, principles, intended uses and data interpretation for each single assay type and their potential applicability for describing and estimating quantitative key event relationships of various NGTxC mechanisms keeping in mind that the remit of this work is to support the pragmatic regulatory applications for predicting cancer hazard and risk. 

### 3.1. Classification Systems and Assays

A variety of cancer marker classification systems and assays were reviewed and are discussed below and summarised in Table 1. 

On the basis of the information provided by the developer, an initial review of the assays was performed with the PANTHER (Protein Analysis Through Evolutionary Relationships) Classification System [33,34]. The system is part of the Gene Ontology Phylogenetic Annotation Project [35] and it allows the classification of proteins (and encoding genes) according to family and subfamily, molecular function, biological process and pathway. They are annotated with Gene Ontology (GO) terms, and sequences are assigned to PANTHER pathways. Thus, in PANTHER, functional classification of each gene is described as: pathway, GO molecular function, GO biological process (i.e., cellular, metabolic, biological regulation), GO cellular component and PANTHER protein class.

The QuantiGene RNA Assay (Thermo Fisher Scientific Inc., Waltham, Massachusetts, US) aims to profile sets of gene expression, with pathway categories developed based upon expert knowledge (Thermo Fisher Scientific Inc., personal communication). Three different QuantiGene panels were analysed with PANTHER: specifically, the cancer pathway; stress & toxicity; and epigenetic chromatin modification enzymes. The cancer pathway panel includes genes involved in a number of pathways, among the most represented are apoptosis, angiogenesis, integrin signalling, inflammation mediated by chemokine and cytokine, p53 and CCKR signalling pathways (Appendix A). For example, the genes included in the cancer pathway panel encode mainly for protein modifying enzymes, cell adhesion molecules, transmembrane signalling receptors and intercellular signal molecules. The most highly represented protein classes are identified by red boxes (Appendix A). The stress and toxicity gene panel includes a selection of genes mainly involved in apoptosis, inflammation mediated by chemokine and cytokine signalling, p53 and CCKR signalling pathways. The most represented protein classes are metabolite interconversion enzymes and nucleic acid metabolism proteins (Appendix A). The epigenetic chromatin modification enzymes panel includes genes encoding protein classes related to chromatin/chromatin-binding, or regulatory proteins, protein modifying enzymes and nucleic acid metabolism proteins. These genes are mostly involved in WNT signalling (Appendix A). 

The Attagene-cis Factorial™ Assay (Attagene Inc., Morrisville, NC, US) aims to assess transcription factor activities across various biological processes [36]. The PANTHER analysis shows the prevalence of gene-specific transcriptional regulators within the cellular, metabolic and biological regulation processes [GO: 0009987, GO: 0008152 and GO: 0065007, respectively]. They function in the apoptosis, angiogenesis, Gonadotropin-releasing hormone receptor, PDGF, CCKR and WNT-signalling pathways (Appendix A). 

The nCounter Pan-Cancer Assay panel (NanoString Technologies Inc., Seattle, DC, US) and the TruSight RNA Pan-Cancer panel (Illumina Inc., San Diego, CA, US) represent two other examples of gene panels which aim to cover multiple carcinogenesis markers. The nCounter Pan-Cancer Assay panel measures gene expression with genes representing all major cancer pathways, from 13 cancer-associated canonical pathways including WNT, Hedgehog, apoptosis, cell cycle, RAS, PI3K, STAT, MAPK, NOTCH, TGF-β, chromatin modification, transcriptional regulation, and DNA damage control (Appendix A). TruSight is based on targeted RNA-seq and includes genes mainly encoding for gene-specific transcriptional regulators and protein modifying enzymes involved in gonadotropin-releasing hormone receptor, angiogenesis, CCKR signalling map, WNT signalling, inflammation mediated by chemokine and cytokine signalling, FGF signalling pathways (Appendix A). 

The BCScreen, a gene panel recently designed by Grashow et al., specifically addresses a number of markers relative to breast carcinogenesis [37]. The PANTHER analysis confirmed the properties of the BCScreen model. As such, apoptosis, gonadotropin-releasing hormone receptor, angiogenesis, CCKR signalling map, p53, interleukin signalling, inflammation mediated by chemokine and cytokine signalling pathways are highly represented and clustered within the cellular, metabolic and biological regulation process as from PANTHER analysis (Appendix A). 

The gene panels of the above-mentioned assay types were further curated using The Human Protein Atlas database (https://www.proteinatlas.org/, accessed on 30 November 2021), which enabled exploration of protein function in the context of the most curated human metabolic network and pathways. The main biological processes identified across the assay types are summarized in Table 1.

In addition to the BCScreen, we also considered the 131 genes with altered expression (e.g., genes with epigenetic silencing, copy number changes) in different types of cancers analysed by TCGA [38] and ~250 relevant molecular targets for the paediatric cancers published by the US FDA (https://www.fda.gov/about-fda/oncology-center-excellence/pediatric-oncology, accessed on 26 January 2021), as reference gene sets that are involved in carcinogenesis. The genes within three panels of assays namely—nCounter Pan-Cancer Assay panel, Quantigene Cancer Pathway Panel and TruSights Pan-Cancer Panel, all of which cover a large number of signalling pathways and biological processes were compared with BCScreen genes, TCGA genes, and the FDA’s paediatric targets. Overlap was as follows; 20.8%, 13.2% and 28.4% of BCScreen genes, 56.5%, 18.3% and 64.1% of TCGA genes and 27.5%, 11.5% and 43.0% of FDA’s paediatric cancer targets. Not surprisingly, these analyses show that a single assay often does not adequately address key signalling pathways or other biological pathways that are potentially affected during carcinogenesis. Whole transcriptome analysis in combination with pathway analysis could provide a promising alternative approach to design and the selection of multiple transcriptomic assay combinations, thereby refining the strategy to assay combination for subsequent follow-up. 

Whilst agnostic assay approaches such as RNA sequencing can identify these pathways also, the use of the pre-prepared and prioritised markers derived in Table 1 can expedite and harmonise the key marker extraction and application to the NGTxC IATA.

### 3.2. Rodent and Human Biomarker in Carcinogenicity Studies

Gene sets are derived to predict for genotoxicity and receptor-mediated toxicity. For example, the use of toxicogenomics-MAPr (TXG-MAPr, https://txg-mapr.eu/, accessed on 10 December 2021) to build such predictive biomarkers, applied to hepatocarcinogens in rodent models and clinical studies for pharmaceuticals, looks promising, as reported in two recent independent studies [39,40]. Corton et al. determined six common MIEs in rodent liver cancer adverse outcome pathways (AOPs) using short-term in vivo assays, for the early identification of carcinogenic potential. These gene sets are predictive of genotoxicity and the activation of one or more xenobiotic receptors, i.e., the AhR, the constitutive androstane receptor (CAR), the oestrogen receptor (ER), and the peroxisome proliferator activated receptor α (PPARα). The sixth biomarker was cytotoxicity, as chronic injury is important in liver tumorigenesis [39]. By using existing transcriptome data from a rat liver microarray compendium with 2013 comparisons of 146 chemicals administered at rat tumorigenic doses, the genes in each biomarker set were characterised through the unsupervised TXG-MAP network model, then combined with the US EPA Toxicological Priority Index (ToxPi) to rank chemicals. Balanced accuracies, evaluating sensitivity and specificity, using thresholds derived from either TG-GATES or DrugMatrix datasets to predict tumorigenicity in independent sets of chemicals, were up to 93%. These results show that a MIE-directed approach using only gene expression biomarkers could be used in short-term assays to identify chemicals that induce tumours, and the doses at which those chemicals induce tumours in rodents. However, further work is required in terms of assessment of the human relevance of these pathways, including the quantitative relationships of the KE’s involved. Moreover, since non-animal approaches would be highly preferred over in vivo studies, translation of this approach to an in vitro setting would be desirable.

Interestingly, Callegaro et al. further developed the application of the TXG-MAPr model web tool (available at https://txg-mapr.eu/WGCNA_PHH/TGGATEs_PHH/, accessed on 9 November 2021), that weights gene co-expression networks (WGCNA) obtained from the Primary Human Hepatocytes (PHH) TG-GATEs dataset [40]. On the basis of analyses of 50 different PHH donors’ responses to a common stressor, tunicamycin, the authors constructed module associations using donors pre-existing disease states/variability. Gene co-expression modules were annotated with functional information (pathway enrichment, transcription factors) to enable mechanistic interpretation. Stress response pathways of heat shock proteins, immune response, mitochondrial response, DNA damage and oxidative stress were captured in the modules, all of which were perturbed by specific stressors and were also preserved in rat liver, highlighting stress responses that translate across species/testing systems. However, while previous studies have shown conserved patterns of transcriptional responses upon chemical exposure in in vitro and in vivo settings, clear differences in responses have also been reported for primary rat hepatocytes versus rat liver [41,42,43,44,45]. Similarly, there are notable differences between cell lines that require consideration for regulatory application (this is discussed further in Section 4.2).

Liver, being a target organ for xenobiotic metabolism, is a tissue commonly examined in in vivo studies; however, the translation of findings in the liver to other tissues may not be so straightforward. Thus, although there are promising early markers for liver cancer, they do not necessarily apply across cancer tissue types. 

For the time being, target tissues such as the mammary gland are not routinely collected in regulatory in vivo short-term or reproductive studies, although the case to do so has been proposed [46,47,48,49], and as of April 2022, a feasibility project to examine how to assess mammary glands, is now on the OECD Test Guideline Programme workplan. An analysis of the intra-tumour genetic alteration in 21 breast cancers by Nik Zainal et al. identified the variety of changes such as MYC, ERBB2 or CCND1 amplification [50]. Differences in the signalling pathways were also observed within the subtypes of triple-negative (i.e., negative for ER, progesterone receptor (PgR) and HER2) breast cancers [51]. These analyses reflected the strong inter-tumour heterogeneity amongst specific types of cancers and suggest that it would be difficult to generalise the early and late gene expression changes for breast cancers. 

### 3.3. Example of a Tool for Pre-Screening the Gene Expression Changes Associated Carcinogenic Phenotypes

As discussed in the Section 3.1, here we focus on the whole transcriptome analysis in combination with pathway analysis. To apply this approach, the MIE and KEs of the signalling pathways considered relevant for the NGTxC IATA [12] were examined in conjunction with the pathway analysis of the BCScreen and the curated cancer signalling pathways by TCGA [37,38]. Perturbations in these latter signalling pathways also depict possible biological processes related to carcinogenesis. As a proof of principle case study, we selected the 74 gene sets for GSEA from the hallmark gene sets and oncogenic signature gene sets of the Molecular Signatures Database (MsigDB, https://www.gsea-msigdb.org/gsea/msigdb/collections.jsp, accessed on 30 September 2021) [26,52] covering all the signalling pathways from the BCScreen and TCGA (Table 2). The genes included in the gene sets are also available in Appendix A. Some of the gene sets are derived from transcriptomic data of specific cancer cell lines or tissues (e.g., MCF7, DLD1, mouse prostate tissue etc.). These gene sets may limit the possibility to screen the precursor signalling steps in tumour formation. However, several studies have identified the signalling pathways affected in normal human tissues, benign tumours or other types of cancers cells/tissues using such gene sets (Table 2). These studies suggest that such gene sets are still useful for the identification of perturbation of signalling pathways in a wide variety of cells and/or tissues in response to chemical exposure.

## 4. Critical Elements to Include When Designing Transcriptomic Testing Combinations

Fundamental elements for all standard cell culture require appropriate quality control measures, including detailed and standardised protocols; adhering to omics reporting standards; setting quality criteria for designing and interpreting transcriptomic testing [23].

To ensure that the transcriptomics assays are fit for purpose for the NGTxC IATA, several additional critical elements need to be considered as summarised in Table 3.

### 4.1. Prioritisation of Cell Types and Cell Lines to Be Used in Transcriptomic Assays

It is prudent to apply transcriptomics approaches to appropriate test systems for analyses of gene expression changes by chemicals. The responses to chemicals differ between cell lines for a great many reasons, most obviously in relation to the tissue and cellular function from which the cell lines were originally derived. Cell lines also vary in their epigenetic status, and culture conditions can alter the epigenetic status [99]. To facilitate the phenotypic and genotypic characterisation of cell lines, proteomic and genomic resources for cell lines are available. For example, the Cell Line Project of Catalogue of Somatic Mutations in Cancer (COSMIC, https://cancer.sanger.ac.uk/cell_lines, accessed on 19 October 2021) provides the mutations in more than 1000 cell lines [100] and a recent study provided the quantitative proteomic landscape of 375 cancer cell lines in the Cancer Cell Line Encyclopaedia (CCLE) [101]. The LINCS database is searchable for datasets about effects of experimental reagents (small molecules, proteins, antibodies and other) on 180 human cell types (mainly cancer cell lines but also some non-cancer cell lines, primary cells, iPSCs, ESCs, differentiated cells) including the characteristics of the cells and assay protocols (https://lincs.hms.harvard.edu/db/, accessed on 19 October 2021). Some resources are available to ease systematic data search within the LINCS database [102], but given the speed of developments in the field, not all of these current resources may survive. Knowing that gene expression profiles vary greatly between different cell lines, it is important to take into account cell-specific gene expression profiles [103]. This has certainly been an important concern for the CTA, the first in vitro transformation mode of action tool proposed to the OECD [30,32,104,105]. All the CTA models offer the advantage to provide a phenotypic anchoring of onco-transformation, however while the Bhas 42 cells are considered to be initiated cells, due to the integration of multiple copies of H-Ras, whereas the Syrian Hamster Cells (SHE) CTA cells are considered not to be pre-initiated. Thus, this system is able to detect the precursor steps to initiation, while the BalbC 3T3 cell line may be considered to be an intermediate stage between these two variants [30], (Colacci et al., paper in preparation). The phenotypic endpoint in the Bhas 42 model is very likely related to the activation of Ras-dependent signal transduction, which plays a fundamental role in the progression from hyperplastic or dysplastic lesions to malignant lesions in human cancer [104,105].

This example of the differences in the CTA models illustrates how important it is to pay close attention to the characterisation of the cells used, in order for the model to be applied appropriately. Looking at an OECD test guideline relevant cell line, the differentiation status of HepaRG cells, tested in proliferation stage or in confluence, is known to influence the expression of metabolizing enzymes and the extent to which they can be induced [106]. Other studies have characterized the variability of transcriptomic responses to various inducers of cell stress with hiPSCs, hiPSC-derived hepatocyte-like cells (hiPSC-HLCs), PHH and the HepG2 cancer cell line [107]. It has also been noted that sensitivity for different types of cell stress varies between cell types. On a data-driven basis, hiPSC-HLCs appear to be more practical compared to PHHs, and to be more metabolically competent than HepG2 cells. Furthermore, hiPSC-HLCs may be suitable for several cellular stress response pathways such as the NF-kB pathway. In contrast, undifferentiated hiPSCs appeared to be most sensitive for TP53 target genes [107]. These and other omics studies provide valuable insights into the criteria needed for the selection of appropriate cell lines for transcriptomic applications (Appendix A). A critical and comparative assessment of the variability and robustness of a gene set in relation to a particular mechanism or pathway is essential.

The OECD Guidance on Good In Vitro Methods Practices (GIVIMP) [108] and further work on Good Cell Culture Practice (GCCP) provides basic orientation for cell type selection and respective documentation needs in terms of origin, identity (including gender), epigenetic/genomic and phenotypic status, stability and functionality including biotransformation capacity, contamination-free culture, ethical aspects and intellectual property and permission(s) for use. It is also important to combine metabolic activation capacity with test systems, which may increase human relevance [109,110].

Beyond the need for documentation of these basic standard cell culture requirements, the summary box below lists some specific considerations that may be addressed in the shorter term for the selection of cells for transcriptomic readouts, to be used in the IATA for NGTxCs. Development needs to address optimum population specific requirements. This will probably take a longer term to achieve, and will support the evolution of the NGTXC IATA for more targeted needs see Box 1. 

Box 1Summary Box: Prioritised considerations for the selection of cells for transcriptomic read outs shall include the cell’s potential.
**Priorities in the short term;**
(a) to provide human relevant data,(b) for replication, to allow progression of carcinogenesis,(c) for phenotypic anchoring of transcriptomic read-outs with late key events,(d) for long term maintenance/immortality (for practical laboratory reasons),(e) to test a healthy, non-tumorigenic, p53 competent status, or an aged status or any spe-cific disease status, depending upon whether the goal is to protect the healthy population and/or an aged or a diseased population.  
**In the medium term;**
(f) to be used within more complex 3D models and/or high-throughput methods,(g) to be used under animal-product free culturing conditions.  
**And in the longer term;**
(h) to generate data for the variability of transcriptomic responses due to human genetic variability,(i) to generate data for the variety of healthy as well as aged human cell types important for carcinogenesis.

For shorter more immediate term relevance, while healthy, human cells may appear most relevant, the current reality is that well-characterized, stable animal cells may be the optimal choice, even if progressed in carcinogenesis to some extent.

Longer-term considerations may be accomplished within a test design for which hiPSCs are derived from a variable population and differentiated into tissue specific cells, then used in a complementary way.

Moreover, the selection of methods for (pre)validation requires considerations beyond the selection of the cell-type. The relevant criteria include the assessment of the potential role of the method within the IATA and are described in [12].

Whilst for target oncology drug development it is highly relevant to identify in vitro models that can faithfully recapitulate some important aspects of specific cancer types as noted for their specific gene expression profiles, in contrast chemical hazard assessment needs in vitro models that represent or approximate healthy cells. However, for both purposes the gene expression profiles may be useful to make decisions on the suitability of a cellular model. For example, the gene expression profiles of breast cancer cell lines have been shown to be correlated to that of primary breast tumours. Gene expression profiles were obtained for the cell lines from the Genomics of Drug Sensitivity in Cancer (GDSC) [111] and CCLE [112] projects and were then analysed to determine the correlation of the gene expressions between these sets of cell lines with TCGA primary tumours. The coefficient correlation was equal to ~0.6 (Figure 2), suggesting that no major differences were observed between the different cell lines at the gene expression level. In similar studies [113,114] the correlation ranged between 0.8 and 0.5; high copy number variation (CNV) accounted for the high score observed by Jiang et al. [113]. Of the cell lines used in xenograft models, luminal cell lines such as MCF7 and T47D form tumours in the presence of oestrogen. HER2 cell lines (i.e., SKRB3 and MDAMB453) have poor tumorigenic potential [113]. However, for the investigations related to luminal breast cancer types, BT483 and T47D luminal cell lines seem to be the most suitable cell lines. Basal-like cell lines show higher correlation with TGCA HER2 and basal primary tumours than with Luminal A-B types. These analyses suggest that the commonly used breast cancer cell lines keep characteristics similar to the primary tumours with respect to gene expression. Such analyses can be useful for the selection of the most appropriate cell line(s) for the application of transcriptomics in assays to be selected for the NGTxC IATA.

### 4.2. Duration of Chemical Exposure and Concentration/Dose Selection

As already indicated, multiple durations of exposure and relevant concentrations should be investigated with transcriptomic assays to maximise the possibility to detect specific mechanisms of action and responses to potential NGTxCs. The integration of phenotypic in vitro assays or short term sub-acute/sub-chronic repeated dose toxicity studies might provide insights with respect to the doses/concentrations and/or durations of exposure to be investigated in the transcriptomic assays [30,115,116,117,118] as an intermediate step to facilitate the transition from in vivo assays to in vitro.

Several literature reports describe how organ-on-a-chip methodology can be applied to improve the translation across from in vitro to in vivo and vice versa. For example, Jiang et al. [119] suggest that a flexible microfluidic platform can be used to bridge the gap between in vitro and in vivo conditions. The standard transcriptomic analysis of these cell systems can be complemented with causality-inferring approaches to improve mechanistic understanding. These approaches involve statistical techniques that can assist the elucidation of gene regulatory interactions for some aspects of the mechanisms. McMullen et al. developed a web-based interactive browser to facilitate visualization of perturbed pathways following population with expression data from TG-GATEs [41]. When evaluating the extent to which gene expression changes from in-life exposures could be associated with modes of action they considered that a similarity index, the Modified Jaccard Index (MJI) which provides a quantitative description of genomic pathway similarity (rather than gene level comparison) was of greater regulatory utility. Some clusters aggregated chemicals with known similar modes of action, including PPARα agonists (median MJI = 0.315) and NSAIDs (median MJI = 0.322). Analysis of paired in vitro (hepatocyte)-in vivo (liver) experiments revealed systematic patterns in the responses of model systems to chemical stress. Accounting for these model-specific, but chemical-independent, differences improved pathway concordance by 36% between in vivo and in vitro models. Luijten et al. similarly utilised TG-GATEs for a comparison approach to transcriptomics data for 137 substances with divergent modes of action for rat primary hepatocytes and rat liver [42]. They report that a relatively small number of matches observed in vitro were also observed in vivo, but a large number of matches between chemicals were found to be relevant either solely in vivo or solely in vitro. While they could not explain this, they conclude that for the relevant chemical matches, the mechanisms perturbed in vitro are consistent with those perturbed in vivo [42].

Klaren et al. investigated the in vitro-to-in vivo concordance of the signalling response profiles of 130 substances [120]. Signalling response profiles were compared between in vitro data produced through Tox21 and short-term (5 days or shorter) rat liver transcriptomic data. An overall average percent of agreement of a global in vitro-to-in vivo comparative analysis of pathway-level responses are 79%, ranging on a per-chemical basis between 41–100%. The concordance amongst inactive chemicals in both in vitro and in vivo was 89% and those amongst chemicals showing in vitro activity was 13%. In this study, several attributes that affect the concordance were identified, i.e., cell type, target pathways, and physical-chemical properties such as log P. Whilst this information is useful to consider for the interpretation of transcriptome data, it should be noted that the quality of the high throughput in vitro data is not always well supported by the wider literature (see for example [121]), and therefore if supporting evidence is lacking for a weight of evidence assessment, these approaches have lower confidence.

In a series of high throughput (HTP) thyroid toxicity studies focused upon the sodium-iodide symporter which mediates the uptake of iodide into the thyroid, Buckalew et al., however conducted a series of in vitro-in vivo experiments that enabled mutual information generation to improve subsequent experimental design and analyses, thereby improving regulatory confidence in the HTP generated data [122].

Therefore, if HTP screening processes use assays that are not well characterised, and where the assay protocols are not available, such that it becomes problematic to reproduce the data, and there is no supporting species relevant in vivo information, a consequence is that the in vitro data generated using these approaches cannot be confidently applied for the NGTxC IATA purposes. Phenotypic analyses followed by transcriptomic assay combinations would be needed for the identification of biological processes affected, and to enhance the potential for regulatory utility.

Finally, with respect to cost implications, it is notable that sequencing approaches such as NGS are progressively reducing in cost, thereby increasing routine application potential, and enabling the studies with multiple doses and duration of exposure. The US National Human Genome Research Institute estimated the cost of whole human genome sequencing was about US $1000 in 2020, as compared to a cost of $10,000 in 2011 [123].

### 4.3. Threshold Development for the Gene Expression Assay

It is a regulatory challenge to reliably identify a point of departure of gene expression changes using transcriptomics tools, and the ideal approach remains unclear for all the approaches being reported thus far. Recent advances in bioinformatic approaches now apply non-parametric and linear methods, Bayesian methods, or counting methods (only applied for the RNA-seq). Analysis of gene expression patterns such as GSEA or other methods utilises expression scores with statistical values such as *p*-value and false discovery rate (FDR) (reviewed in [124]). A large number of microarray and RNA-seq data are publicly available in databases such as Gene Expression Omnibus (GEO, https://www.ncbi.nlm.nih.gov/gds/, accessed on 1 October 2021) [125] or ArrayExpress (https://www.ebi.ac.uk/arrayexpress/, accessed on 1 October 2021) [126]. Combining more complex statistical methods with the data from these databases may provide a useful resource to determine relevant thresholds for the progression through the KEs, identifying where the shifts from adaptive towards the adverse outcome are being triggered, and this will be particularly valuable for the purposes of the NGTxC IATA.

### 4.4. Tools to Identify Potential Reference Chemicals

For regulatory applications, it is essential that test methods including the technology platforms, software and their application to biological systems, are shown to be reproducible [127]. Reference chemicals should be representative of the range of responses and effects that the validated test method is capable of measuring or predicting consistently, with a good portion of negative chemicals, and reflect the accuracy of the validated test method. Several chemicals including molecular targeting drugs activate specific pathways, and these may be useful as initial reference chemicals. For example, N-(3-Benzylthiazol-2(3H)-ylidene)-1H-pyrrolo[2,3-b]pyridine-3-carboxamide (also known as Lats-IN-1) inactivates the Hippo pathway through the inhibition of LATS1/2 kinase [128]. Vemurafenib, dabrafenib and encorafenib activate CRAF kinase via positive feedback by the inhibition of BRAF kinase [129]. Whilst often more is known about pharmaceuticals as potential reference chemicals, from (therapeutic and safety assessment) cancer research efforts, chemical selection also needs to cover other industrial chemical sectors, for the purpose of the OECD NGTxC IATA.

As the field of toxicogenomics has progressively advanced over the last decades, the knowledge of gene expression changes triggered by chemical exposure has grown. Such knowledge compilation is exemplified by the comparative toxicogenomics database (CTD, http://ctdbase.org/, accessed on 1 October 2021) [130], Open Toxicogenomics Project-Genomics Assisted Toxicity Evaluation system (TG-GATEs, https://toxico.nibiohn.go.jp/english/, accessed on 1 October 2021) [131]. Connectivity Map (C-MAP, https://clue.io/cmap, accessed on 1 October 2021) [132,133], and pattern matching algorisms of differential gene expression and chemicals can also help in the identification of possible reference chemicals in silico by querying the gene expression sets of genetic perturbation of specific pathways. In addition, database searches for the genes responsible for each biological process in combination with the above mentioned toxicogenomics databases may identify the reference chemicals to be used for the case studies and to validate the potential transcriptomic method.

In addition to the available databases, Desaulniers et al. reviewed the effects of several chemicals affecting the various signalling pathways and biological processes through the epigenetic mechanisms [99]. Vaccari et al. also reviewed the chemicals impacting upon cellular senescence (Vaccari et al. In preparation). Rudel et al. conducted a literature search to find human evidence for chemicals that may cause breast cancer, resulting in the identification of 67 chemicals [134]. We also manually curated the genes identified in the literature as playing a role in breast cancers (Appendix A) and the chemicals affecting these gene pathways as a case study (Table 4).

As another case study exercise, we extracted genes that were differentially activated in the breast cancers by using VIPER, an algorithm to infer protein activity using regulon analysis [135] (Table 5). We extracted 141 significant gene pathways found by using gene regulatory network (FDR < 0.05) (Appendix A). A total of 24 chemicals overlapped between two analyses.

These analyses may also help to identify potential reference chemicals for the specific gene expression or pathways indicated, for subsequent next testing steps within the IATA.

### 4.5. Proposed Key Omics Markers for Inclusion in the NGTxC IATA

Extrapolating and consolidating from the analyses conducted upon the data sets described and analysed herein, we have identified several key transcriptomic markers related to carcinogenesis that can be specifically mapped to the early key events of inflammation, immune response, mitogenic signalling and cell injury in the NGTxC IATA [12], as shown in Figure 3. Many of these markers are also clearly identified in the in vitro test methods that can assess the next IATA step after sustained proliferation (as discussed in Section 2).

Given that a number of biological processes needs to be examined within the IATA, test methods that enable the initial screening of such changes can help with prioritisation of subsequent more complex assays within the IATA decision-making process. Most relevant transcriptomic-based approaches that could be utilised as pre-screening tools can be elucidated using the approaches described herein. We have shown how confidence can be increased when using multiple database and experimental approaches in order to pinpoint and cross-confirm the more robust transcriptomic markers that regulators will be able to utilise and industry will be able to report, with greater confidence. Figure 3 includes the most promising of the gene signalling markers that we identified. These can then be utilised for the assays or test methods being evaluated for the key events of the IATA in parallel work, as recently reported for epigenetic DNA methylation [30], gap junction [32], cell transformation [30,32,99,136], and similar work currently underway such as that for immune dysfunction (Corsini et al., paper in preparation), etc. 

## 5. Conclusions

Here we have reviewed and examined how the omics tools and resources that are currently available can be utilised to pre-screen the changes in cell signalling pathways possibly leading to carcinogenic phenotypic outcome during the chemical hazard assessment of NGTxCs. With a focus on chemical regulatory hazard assessment, we have discussed the requirement for robustness and complementarity of relevant mechanistic data sets leading to key events that can lead to carcinogenic phenotypic outcomes. With the need for weight of evidence mutually supportive data sets, we have given examples as to how this can be approached, to start to demonstrate how omics information could be submitted to regulatory authorities, and how those authorities might consider the assessment of such data.

In some cases, the data collection and reporting can be regarded as relatively straight forward, but there are several aspects that require further elucidation and discussion. For example, with respect to continuous versus discrete modelling, it remains to be determined, if the choice of the critical effect size of e.g., 10% or 50% could affect the BMD to a critical extent, e.g., such that the final limit value including its confidence interval (derived by quantitative in vitro to in vivo extrapolation (QIVIVE) modelling) is significantly altered.

Highly standardised workflows are needed for regulators so that they can utilise the data effectively, for their assessments, and here we do provide clear lists that can be utilised for such a purpose (Figure 3). Indeed, using a dedicated gene set/whole transcriptome/gene set that report on transcriptomics data perturbation predictive for NGTxC will go some way to overcome the challenges for industry with regard to their concerns as to how their submitted transcriptome data might be utilised at later dates.

A data-reporting framework will be needed, and this is in progress at OECD level, for both the reporting format for the collection and the processing of the data [23] together with a more defined regulatory framework describing how the data streams can be optimally interpreted and trigger subsequent or parallel testing modules, in a transparent fashion.

In the longer term, it will be helpful to examine the potential benefits from the use of more complex 3D systems as compared to 2D cell lines. A recent survey indicated that regulators have high interest in establishing assessment approaches for complex models, such as 3D cultures, spheroids, microphysiological systems including organ-on-chip devices, bioreactor cultures or bioprinted tissues, but these assessment procedures still need to be established [137]. Indeed, in the field of genotoxic carcinogenicity 3D skin models were reported to “allow for more natural cell–cell and cell–matrix interactions, and show ‘in vivo-like’ behaviour for key parameters, such as cell proliferation, differentiation, morphology, gene and protein expression, and function” and the false positive rate of in vitro mutagenicity or chromosome aberration tests is claimed to be reduced [138]. However, new 2D systems appear to perform better than the current genotoxicity methods, such as the cell-line (Huh6) with greater metabolic competence when used within a micronucleus test [139] or mouse stem cells used within the Toxtracker transcriptomics marker approach [140]. Thus, simpler higher throughput models could be sufficient now for regulatory use for classification and potency assessment purposes, while more complex models may substitute the simpler models in some cases in the future, when mature and validated, and their specific added value is clearly demonstrated.

We acknowledge the challenges and complexity of selecting proper gene panels, cell types and culture systems, test concentrations, analysis time-points and assessment procedures for transcriptomic data. However, this is not necessarily an obstacle for the short- term use of transcriptomics data in the context of the envisaged IATA for NGTxC. This is because testing for signalling pathways is not considered a stand-alone approach. Instead, signalling pathways shall be allocated to each MIE/KE in the IATA (Figure 3) and anchored to phenotypic endpoint assay results for mutual strengthening of the weight of evidence (Figure 1). Within this design perspective, the higher throughput transcriptomics methods may be used as a first tier to target appropriate second tier testing with phenotypic endpoint assays. In addition, the phenotypic assays may be mechanistically characterized by integrating transcriptomics read-outs for their validation and/or during their ultimate regulatory use, and here the application of omics tools to refine such test methods show promise in reducing any uncertainties.

## Figures and Tables

**Figure 1 ijms-23-12718-f001:**
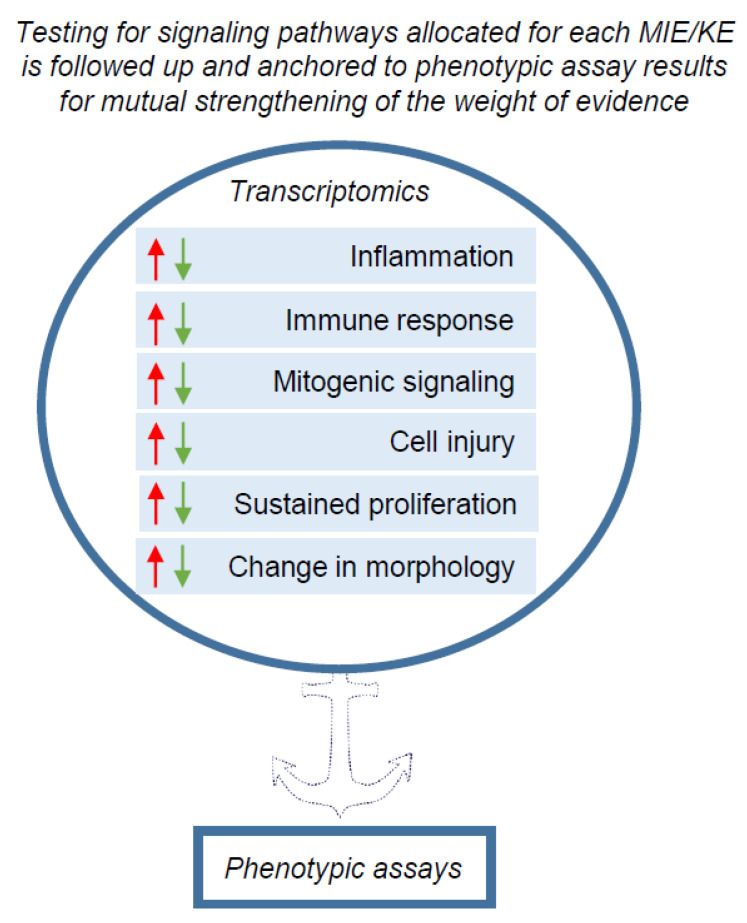
An overview scheme of transcriptomics application for pre-screening the possible phenotypic changes leading to carcinogenesis in the NGTxC IATA. The signalling pathways associated with biological processes for MIE and KEs (boxes with arrow heads reflecting down/up regulation of gene expression) encircled in blue, may be screened by transcriptomic tools. Transcriptomics captures the signalling pathways and ‘flags’ the possible changes in biological processes that can be picked up in more complex in vitro and in vivo assays, particularly with respect to the KE of uncontrolled proliferation for example. The next step is that the phenotypic assays within each KE block/mode of action can further target and delineate the changes in the carcinogenic phenotypes, induced by chemicals.

**Figure 2 ijms-23-12718-f002:**
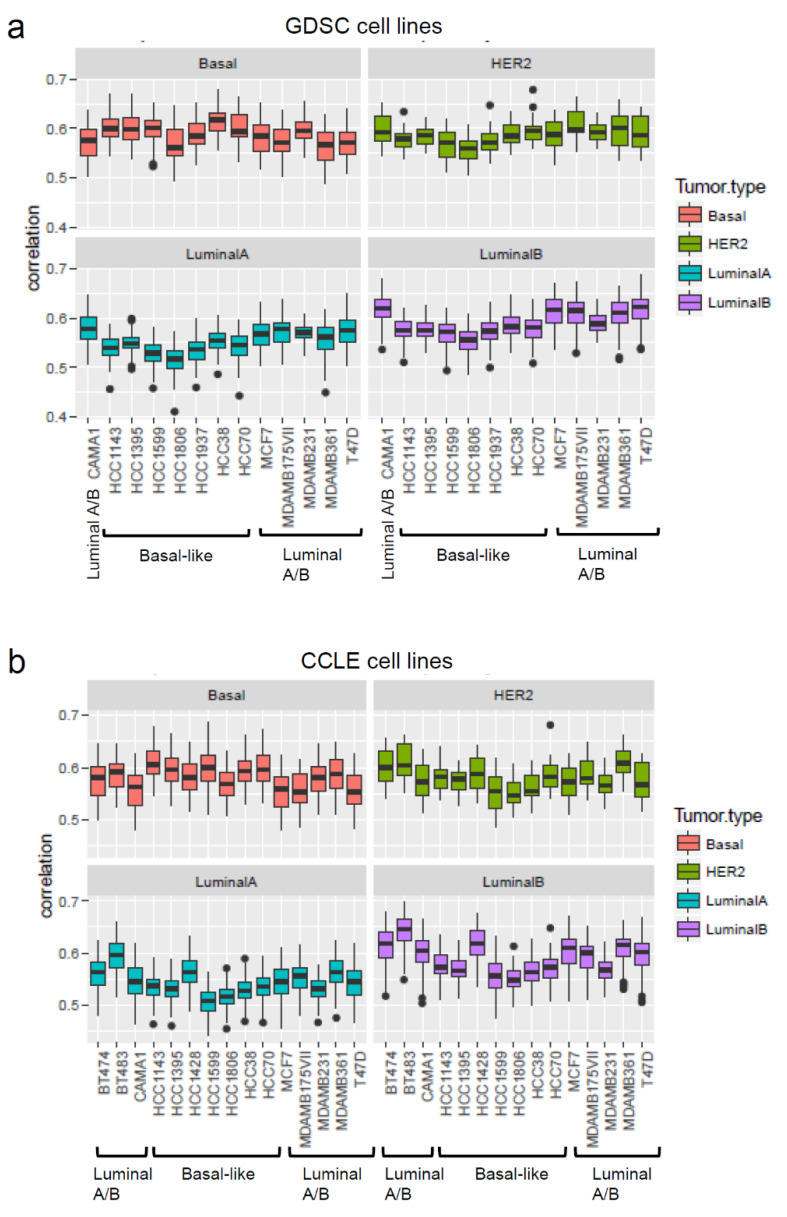
Correlation of gene expressions of commonly used breast cancer cell lines with primary breast tumours. The correlation of the gene expressions of GDSC and CCLE breast cancer cell lines were analysed with primary tumours from TGCA breast cancer analysis. (**a**) GDSC cell lines, (**b**) CCLE cell lines.

**Figure 3 ijms-23-12718-f003:**
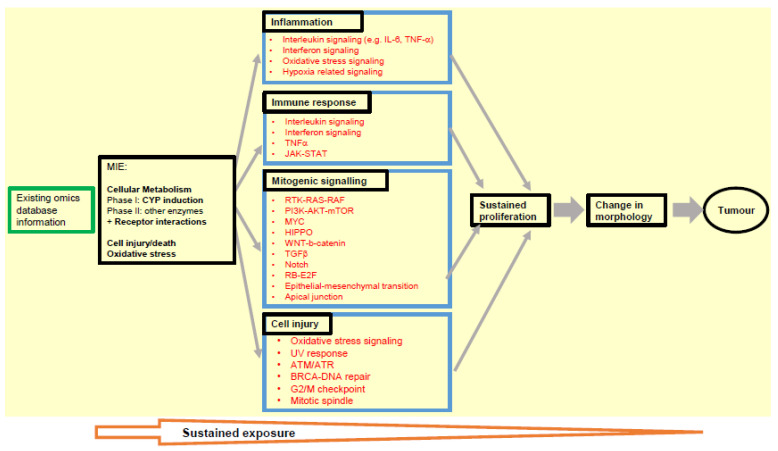
Pivotal omics markers proposed to be monitored in assay tools that address the key events of inflammation, immune response, mitogenic signalling and cell injury, in the NGTxC IATA (yellow section). Signalling pathways contributing carcinogenesis (red) were allocated to MIE and KEs in the IATA. Aberrant regulation of these signalling pathways should be covered by the transcriptomics to identify the possible phenotypic changes, and examples of useful tools and databases for the early key event pre-screening are provided in the main text.

**Table 1 ijms-23-12718-t001:** Comparison of the biological processes addressed by selected transcriptional assays and gene panels.

	Quantigene			Attagene cis-Factorial (n = 83)	nCounter Pan Cancer (n = 767)	Trusight Pan Cancer (n = 1388)	BCScreen Method (n = 500)
	Cancer Pathway (n = 85)	Stress &Toxicity (n = 84)	Epigenetic Chromatin Modification Enzymes (n = 44)				
Biological process:							
cell adhesion		√		√		√	
angiogenesis	√	√		√		√	√
apoptosis	√	√	√		√	√	√
cell cycle	√	√	√	√	√	√	√
development	√			√	√	√	
differentiation	√	√		√	√	√	
DNA repair		√	√		√		
epigenetic alteration			√	√	√		√
genotoxicity	√	√	√	√			√
growth	√	√	√	√	√	√	√
hormone alteration	√		√	√	√	√	√
immortalization							√
immune response	√		√	√		√	√
inflammatory response	√			√		√	√
mammary gland related	√					√	√
proliferation	√			√	√	√	√
oxidative stress response		√	√	√	√	√	√
transcriptional misregulation					√	√	
tumour invasion	√	√	√			√	
tumour suppression	√	√				√	
xenobiotic metabolism		√		√		√	√

Biological processes represented within each assay type. The processes, were manually curated using The Human Protein Atlas Database (https://www.proteinatlas.org/, accessed on 30 November 2021) (n = number of genes per panel).

**Table 2 ijms-23-12718-t002:** Possible gene sets for GSEA applied to transcriptomics assays in the IATA on NGTxCs.

Gene Sets	Descriptions of Gene Sets	Covering Biological Processes	Pathways	Cells/Tissue Used & References
HALLMARK_ XENOBIOTIC_ METABOLISM	Genes encoding proteins involved in processing of drugs and other xenobiotics.	P450 induction	xenobiotic metabolism	Liver cancer [53,54]
HALLMARK_ ANDROGEN_ RESPONSE	Genes defining response to androgens.	P450 induction, receptor binding, transactivation, human receptor	androgen	Prostate cancer [55]
HALLMARK_ OESTROGEN_ RESPONSE_EARLY	Genes defining early response to oestrogen.	P450 induction, receptor binding, transactivation, human receptor	oestrogen	Prostate cancer [56]
HALLMARK_ OESTROGEN_RESPONSE_LATE	Genes defining late response to oestrogen.	P450 induction, receptor binding, transactivation, human receptor	oestrogen	Prostate cancer [56]
RELA_DN.V1_DN RELA_DN.V1_UP	Genes down/up-regulated in HEK293 cells (kidney fibroblasts) upon knockdown of RELA gene by RNAi.	Immunoevasion, immunotoxicity, inflammation	NFκB	Breast cancer [57] Hepatocellular carcinoma [58]
HALLMARK_IL6_ JAK_STAT3_ SIGNALING	Genes up-regulated by IL6 via STAT3, e.g., during acute phase response.	Immunoevasion, immunotoxicity, inflammation	IL-6	Osteosarcoma [59]
HALLMARK_ INFLAMMATORY_ RESPONSE	Genes defining inflammatory response.	Immunoevasion, immunotoxicity, inflammation	inflammation	Osteosarcoma [59]
HALLMARK_ INTERFERON_ ALPHA_RESPONSE	Genes up-regulated in response to alpha interferon proteins.	Immunoevasion, immunotoxicity	Interferon α	Osteosarcoma [59]
HALLMARK_ INTERFERON_ GAMMA_RESPONSE	Genes up-regulated in response to IFNG.	Immunoevasion, immunotoxicity	interferon γ	Osteosarcoma [59]
IL15_UP.V1_DN IL15_UP.V1_UP	Genes down/up-regulated in Sez-4 cells (T lymphocyte) that were first starved of IL2 and then stimulated with IL15.	Immunoevasion, immunotoxicity	IL15	Breast cancer [60] Oesophageal carcinoma [61]
IL21_UP.V1_DN IL21_UP.V1_UP	Genes down/up-regulated in Sez-4 cells (T lymphocyte) that were first starved of IL2 and then stimulated with IL21.	Immunoevasion, immunotoxicity	IL21	U-2932 diffuse large B cell lymphoma cell line [62]
IL2_UP.V1_DN IL2_UP.V1_UP	Genes down/up-regulated in Sez-4 cells (T lymphocyte) that were first starved of IL2 and then stimulated with IL2.	Immunoevasion, immunotoxicity	IL2	U-2932 diffuse large B cell lymphoma cell line [62]
JAK2_DN.V1_DN JAK2_DN.V1_UP	Genes down/up-regulated in HEL cells (erythroleukaemia) after knockdown of JAK2 gene by RNAi.	Immunoevasion, immunotoxicity, inflammation	JAK-STAT	U-2932 diffuse large B cell lymphoma cell line [62]
HALLMARK_ REACTIVE_ OXYGEN_ SPECIES_PATHWAY	Genes up-regulated by reactive oxygen species (ROS).	Oxidative stress	oxidative stress response	HT1080 human fibrosarcoma [63]
NFE2L2.V2	Genes down-regulated in MEF cells (embryonic fibroblasts) after knockout of NFE2L2 gene.	Oxidative stress, senescence	NRF2-KEAP1	A549 Lung cancer cell line [64]
HALLMARK_ HYPOXIA	Genes up-regulated in response to low oxygen levels (hypoxia).	Angiogenesis	hypoxia response	Glioblastoma [65]
EGFR_UP.V1_DN EGFR_UP.V1_UP	Genes down/up-regulated in MCF7 cells (breast cancer) positive for ESR1 and engineered to express ligand-activatable EGFR.	Cell proliferation, cell transformation	RTK-RAS-RAF	Pulmonary arcinoids [66] Ameloblastoma [67]
ERBB2_UP.V1_DN ERBB2_UP.V1_UP	Genes down/up-regulated in MCF7 cells (breast cancer) positive for ESR1 and engineered to express ligand-activatable ERBB2.	Cell proliferation, cell transformation	RTK-RAS-RAF	Gastric cancer [68] Huh7 Hepatocellular carcinoma cell line [69]
KRAS.600_UP.V1_DN KRAS.600_UP.V1_UP	Genes down/up-regulated in four lineages of epithelial cell lines over-expressing an oncogenic form of KRAS gene.	Cell proliferation, cell transformation	RTK-RAS-RAF	Colorectal cancer [70] Ameloblastoma [67]
RAF_UP.V1_DN RAF_UP.V1_UP	Genes down/up-regulated in MCF7 cells (breast cancer) positive for ESR1 MCF7 cells (breast cancer) stably over-expressing constitutively active RAF1 gene.	Cell proliferation, cell transformation	RTK-RAS-RAF	Neuroblastoma [71]
MEK_UP.V1_DN MEK_UP.V1_UP	Genes down/up-regulated in MCF7 cells (breast cancer) positive for ESR1 MCF7 cells (breast cancer) stably over-expressing constitutively active gene.	Cell proliferation, cell transformation	RTK-RAS-RAF	Neuroblastoma [71] T cell leukemia cell linnes [72]
AKT_UP.V1_DN AKT_UP.V1_UP	Genes down/up-regulated in mouse prostate by transgenic expression of human AKT1 gene vs controls.	Cell proliferation, cell transformation	PI3K-AKT-mTOR	Bladder cancer [73] Hepatocellular carcinoma [58]
PTEN_DN.V1_DN PTEN_DN.V1_UP	Genes down/up-regulated upon knockdown of PTEN by RNAi.	Cell proliferation, cell transformation	PI3K-AKT-mTOR	Small cell lung cancer [74]
MTOR_UP.V1_DN MTOR_UP.V1_UP	Genes down/up-regulated by everolimus in prostate tissue.	Cell proliferation, cell transformation	PI3K-AKT-mTOR	Keratinocytes/ fibroblast [75] T cell leukaemia cell lines [72]
MYC_UP.V1_DN MYC_UP.V1_UP	Genes down/up-regulated in primary epithelial breast cancer cell culture over-expressing MYC gene.	Cell proliferation, cell transformation	MYC	Hepatocellular carcinoma [58] Colorectal cancer [76]
YAP1_DN YAP1_UP	Genes down/up-regulated in MCF10A cells (breast cancer) over-expressing YAP1 gene.	Cell proliferation, cell transformation	Hippo	Hepatocellular carcinoma [58] Prostate cancer [77]
WNT_UP.V1_DN WNT_UP.V1_UP	Genes down/up-regulated in C57MG cells (mammary epithelium) by over-expression of WNT1 gene.	Cell proliferation, cell transformation	WNT-β-catenin	Peripheral nerve sheath tumour [78] Hepatocellular carcinoma [58]
BCAT.100_UP.V1_DN BCAT.100_UP.V1_UP	Genes down/up-regulated in HEK293 cells (kidney fibroblasts) expressing constitutively active form of CTNNB1 gene.	Cell proliferation, cell transformation	WNT-β-catenin	Hepatocellular carcinoma [58] Acute lympho- blastic leukaemia [79]
LEF1_UP.V1_DN LEF1_UP.V1_UP	Genes down/up-regulated in DLD1 cells (colon carcinoma) over-expressing LEF1.	Cell proliferation, cell transformation	WNT-β-catenin	Bladder cancer [73] Hepatocellular carcinoma [58]
TGFB_UP.V1_DN TGFB_UP.V1_UP	Genes down/up-regulated in a panel of epithelial cell lines by TGFB1.	Cell proliferation, cell transformation	TGF-β	T cell leukaemia cell lines [72] Breast cancer cell lines [80]
NOTCH_DN.V1_DN NOTCH_DN.V1_UP	Genes down/up-regulated in MOLT4 cells (T-ALL) by DAPT, an inhibitor of NOTCH signaling pathway.	Cell proliferation, cell transformation	Notch	Endometrial cancer [81] Oesophageal carcinoma [61]
E2F1_UP.V1_DN E2F1_UP.V1_UP	Genes down/up-regulated in mouse fibroblasts over-expressing E2F1 gene.	Cell proliferation, cell transformation, senescence	Rb-E2F	Acute myeloid leukaemia [82] Prostate cancer [83]
RB_DN.V1_DN RB_DN.V1_UP	Genes down/up-regulated in primary keratinocytes from RB1 skin specific knockout mice.	Cell proliferation, cell transformation, senescence	Rb-E2F	Hepatocellular carcinoma [58] Pancreatic cancer [84]
HALLMARK_ EPITHELIAL_ MESENCHYMAL_ TRANSITION	Genes defining epithelial-mesenchymal transition, as in wound healing, fibrosis and metastasis.	Cell proliferation, loss of gap junction	epithelial-mesenchymal transition	Pan-cancer [85]
HALLMARK_ UV_RESPONSE_DN HALLMARK_ UV_RESPONSE_UP	Genes down/up-regulated in response to ultraviolet (UV) radiation.	Genetic instability, senescence	UV response	Bone marrow stromal cell [86] Prostate cancer [87]
ATM_DN.V1_DN ATM_DN.V1_UP	Genes down/up-regulated in HEK293 cells (kidney fibroblasts) upon knockdown of ATM gene by RNAi.	Genetic instability, senescence	ATM-ATR	Hepatocellular carcinoma [58] Endometrial cancer [81]
BRCA1_DN.V1_DN BRCA1_DN.V1_UP	Genes down/up-regulated in MCF10A cells (breast cancer) upon knockdown of BRCA1 gene by RNAi.	Genetic instability, senescence	BRCA	Hodgkin lymphoma [88] Hepatic stellate cell [89]
HALLMARK_ DNA_REPAIR	Genes involved in DNA repair.	Genetic instability, senescence	DNA repair	Hepatocellular carcinoma [90]
HALLMARK_ G2M_CHECKPOINT	Genes involved in the G2/M checkpoint, as in progression through the cell division cycle.	Cell proliferation, cellular transformation, senescence	G2/M checkpoint	Hepatocellular carcinoma [91]
HALLMARK_ MITOTIC_SPINDLE	Genes important for mitotic spindle assembly.	Cell proliferation, cellular transformation, senescence	mitotic spindle	Breast cancer [92]
CYCLIN_D1_UP.V1_ DN CYCLIN_D1_UP.V1_ UP	Genes down/up-regulated in MCF7 cells (breast cancer) over-expressing CCND1 gene.	Cell proliferation, cell transformation	Cyclin-CDK	Hepatocellular carcinoma [58] Colorectal cancer [93]
P53_DN.V1_DN P53_DN.V1_UP	Genes down-regulated in NCI60 panel of cell lines with mutated TP53.	Cell proliferation, cell transformation, genetic instability, senescence, apoptosis	p53	Colon Adenocarcinoma [94]
HALLMARK_ APOPTOSIS	Genes mediating programmed cell death (apoptosis) by activation of caspases.	Apoptosis	apoptotic pathways	Colorectal cancer [95]
HALLMARK_ APICAL_JUNCTION	Genes encoding components of apical junction complex.	Cell proliferation, loss of gap junction	apical junction, epithelial-mesenchymal transition	Colorectal cancer [96]
VEGF_A_UP.V1_DN VEGF_A_UP.V1_UP	Genes down/up-regulated in HUVEC cells (endothelium) by treatment with VEGFA.	Angiogenesis	angiogenesis	Head and neck squamous cell carcinoma [97] Breast cancer [57]
HALLMARK_ ANGIOGENESIS	Genes up-regulated during formation of blood vessels (angiogenesis).	Angiogenesis	angiogenesis	Breast cancer [98]

**Table 3 ijms-23-12718-t003:** Critical elements to include when designing transcriptomic testing combinations.

Use of human-relevant cell lines for the transcriptomic assays
2. The duration of exposure and concentration/dose selection of the chemicals
Concentration- and time-response relationships to allow for the evaluation of early and late-stage mechanisms involved in carcinogenesis
3. Procedure for point-of-departure derivation
Relevant and robust statistical analyses Consensus on relevant critical effect sizes
4. Reference chemicals and case studies
Use of known carcinogens and non-carcinogens Use of transcriptomic databases and the investigation of mechanisms and modes of action utilising bioinformatics tools

**Table 4 ijms-23-12718-t004:** Chemicals interacting with 60 genes differentially expressed in breast cancers.

Chemicals	Number of Interactions
Estradiol	46
Cyclosporin	45
Benzo[a]pyrene	40
Valproic Acid	39
Calcitriol	39
Tretinoic	37
Coumestrol	37
Tetrachlorodibenzodioxin	36
Copper Sulfate	35
Genistein	34
Cobalt Chloride	34
Resveratrol	33
Acetaminophen	33
7,8-Dihydro-7,8-dihydroxybenzo9apyrene 9,10-oxide	33
Nickel	32
Testosterone	32
Bisphenol A	32
Alfatoxin B1	31
(+)-JQ1	28
Sodium arsenite	26
Fluorouracil	24
Cadmium chloride	22
Mustard gas	22
Decitabine	21
Propionaldehyde	21
Dasatinib	21
Tunicamycin	19
Lucanthrone	19
ICG 001	19
Methotrexate	19
Zoledronic acid	18
Polychlorinated biphenyls	18
Thapsigargin	17
Palbocivlib	16
K 7174	16
Irinotecan	15
β-methylcholine	14
Cupric acid	13
Vinblastine	10

**Table 5 ijms-23-12718-t005:** Top 50 chemicals interacting with 141 genes differentially activated in breast cancers.

Chemicals	Number of Interactions
Valproic acid	77
Cyclosporine	72
Estradiol	62
Benzo[a]pyrene	61
7,8-Dihydro-7,8-dihydroxybenzo[a]pyrene 9,10-oxide	59
Tretinoin	56
Copper sulfate	56
Calcitriol	55
Aflatoxin B1	53
Acetaminophen	53
Nickel	51
Testosterone	50
Coumestrol	50
(+)-JQ1	48
Troglitazone	46
Tetrachlorodibenzodioxin	45
Cobalt chloride	43
Trichostatin A	39
Arsenic trioxide	39
Quercetin	38
Resveratrol	37
Bisphenol A	37
Genistein	36
Hydrogen peroxide	35
Silicon dioxide	33
Formaldehyde	33
Cisplatin	33
ICG 001	30
Fluorouracil	30
Methyl methanesulfonate	28
K 7174	28
Dasatinib	27
Potassium chromate (VI)	27
Phenylmercuric acetate	27
Mustard gas	27
Cadmium	27
Vitamine K3	26
Tert-butylhydroperoxide	26
Propionaldehyde	26
Methotrexate	26
Doxorubicine	26
Zoledronic acid	25
Thapsigargin	25
Decitabine	25
Cadmium chloride	25
Sodium arsenite	24
Carbamazepine	24

## Data Availability

Not applicable.

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
