# Peer review of "Analyses of Transcriptomics Cell Signalling for Pre-Screening Applications in the Integrated Approach for Testing and Assessment of Non-Genotoxic Carcinogens"

_ijms, 2022, doi:10.3390/ijms232112718_

Round 1

Reviewer 1 Report

It is well designed and performed Ames and in vitro micronucleus tests to evaluate commonly used pesticides at field concentrations.

It just needs whole-manuscript English proofread for the best manuscript.

Author Response

Thank you for the comment. Our response is below in red

It is well designed and performed Ames and in vitro micronucleus tests to evaluate commonly used pesticides at field concentrations.

Thank you for this comment, but this manuscript is not about ‘designing and evaluating Ames and in vitro micro nucleus tests to evaluate commonly used pesticides at field concentrations’.

We wonder if this review belongs to the paper by Oku et al that was submitted? The revision is attached.

This is an evaluative review on the use of transcriptomics pre-screening in the IATA for NGTxCs.

Reviewer 2 considers that it is well written.

It just needs whole-manuscript English proofread for the best manuscript.

The English has been proofread by native speaker co-authors, and checked again following revisions as suggested by reviewers 2 and 3.

Reviewer 2 Report

The paper by Oku et al brings a comprehensive overview of the applicability of transcriptomics pre-screening in the IATA for NGTxCs. I think this paper is well written and I have just a few minor comments:

1. I am missing an explanation of how the reported transcriptomic studies relevant for NGTxCs were selected, and based on which criteria.

2. Table 3 - I would also mention including all relevant quality controls, following standardized protocols, and setting quality criteria for designing transcriptomic testing to increase its reproducibility.

3. It is not clear if Figure 2 was made by the authors of this paper or taken from a publication or report. It should be clarified and relevant references should be added to the legend.

4. The tables in the Supplement do not match their description in the main text. For example, Table S2 includes genes differentially expressed in breast cancer for selecting interacting chemicals but not Criteria for the cell line selection.

Author Response

Thank you for your positive comments. 

Our responses to your comments were below in red.

  1. I am missing an explanation of how the reported transcriptomic studies relevant for NGTxCs were selected, and based on which criteria.

We selected commercially or publicly available transcriptomics-based multiplex assays for the evaluation. To reflect this, we revised the manuscript (line 154-158).

  1. Table 3 - I would also mention including all relevant quality controls, following standardized protocols, and setting quality criteria for designing transcriptomic testing to increase its reproducibility.

Thank you. This is a very valuable point, and we have inserted in the main text (line 339-344).

  1. It is not clear if Figure 2 was made by the authors of this paper or taken from a publication or report. It should be clarified and relevant references should be added to the legend.

Figure 2 is original data, not from another publication. We described the analysis that we report in this manuscript (line 447-454). On re-reading, we also corrected the labelling of a cell line. Accordingly Figure 2 is now replaced in the revised manuscript.

  1. The tables in the Supplement do not match their description in the main text. For example, Table S2 includes genes differentially expressed in breast cancer for selecting interacting chemicals but not Criteria for the cell line selection.

Apologies. There was an inconsistency in the numbering of the Supplementary Tables. We have corrected the supplementary table numbers.

Reviewer 3 Report

The authors provide a nice overview how to use transcriptomic data and tools within the IATA to screen for nongenotoxic carcinogens. The paper presents and discusses resources, tools and gene panels currently available for transcriptomic assays relevant for IATA on nongenotoxic carcinogens. It also focuses on the critical elements in the design of transcriptomic studies to test for nongenotoxic carcinogens, and what are the currently available tools, resources and approaches to address these elements in carcinogenicity testing. The paper is well organized and clearly written, and I recommend it for publication. I have only few minor comments:

 Lines 175-215 … Numbers of evaluated genes are listed for some assays/panels, for some not, I suggest to make it consistent.

 Lines 175-215 … numbering of Supplementary Figures is out of order (Fig S5 shall be TruSight, but it is listed as S6 in SM; nCounter shall be Fig S6, but it is listed as S7; BCScreen shall be Fig S7, but it is listed as S5)

 Line 211 and 216 … “Panther” shall be capitalized

 Line 226 … “gene sets that involved in carcinogenesis” – probably “gene sets involved in carcinogenesis / gene sets that are involved in carcinogenesis” ?

 Line 303 … “we selected the gene sets for GSEA” – I suggest to specify the number of gene sets

 Line 308 … “The genes included in the gene sets are also available in Table S1.” – Table S1 does not include/list genes for “HALLMARK_APOPTOSIS”. Also, “HALLMARK_XENOBIOTIC_METABOLISM” appears twice in the Table 2 (Liver – Ref#53 and Breast Cancer – Ref#98) – were these the same sets of genes listed in Table S2 / column A?

 Line 430 … please, add reference number (Ref#113?) after “Jiang et al.”

 Line 437 … please, add reference number (Ref#42?) after “rat liver.”

 Line 491 … define “HTP” – high throughput?

 Line 492 … “Buckalew et al.” - missing dot

 Table S1 … Heading of the Table is placed in the cell BL1, not A1

 Table S2 … (Table S2: 60 genes differentially expressed in the breast cancer for selecting interacting chemicals) - this should be Table S3

 Table S3 … (“cells/cell lines”) - this should be Table S2

Author Response

Thank you for your positive comments. 

Our responses to your comments were below in red.

 Lines 175-215 … Numbers of evaluated genes are listed for some assays/panels, for some not, I suggest to make it consistent.

The numbers of evaluated genes for assay and panels are listed in the Table. Therefore the gene numbers were deleted from the text to make it consistent.

Lines 175-215 … numbering of Supplementary Figures is out of order (Fig S5 shall be TruSight, but it is listed as S6 in SM; nCounter shall be Fig S6, but it is listed as S7; BCScreen shall be Fig S7, but it is listed as S5)

There was an inconsistency between figures and the figure legend in supplementary figures. We corrected the numbering, it is aligned with Table 1. We also ordered the text describing the assays accordingly (line 180-225).

 Line 211 and 216 … “Panther” shall be capitalized

According to the reviewer’s comment, “Panther” now in line 227 and 232 are capitalized.

 Line 226 … “gene sets that involved in carcinogenesis” – probably “gene sets involved in carcinogenesis / gene sets that are involved in carcinogenesis” ?

According to the reviewer’s comment, this part was revised to “gene sets that are involved in carcinogenesis”.

 Line 303 … “we selected the gene sets for GSEA” – I suggest to specify the number of gene sets

According to the reviewer’s comment, we added the number of gene sets (74) here.

 Line 308 … “The genes included in the gene sets are also available in Table S1.” – Table S1 does not include/list genes for “HALLMARK_APOPTOSIS”. Also, “HALLMARK_XENOBIOTIC_METABOLISM” appears twice in the Table 2 (Liver – Ref#53 and Breast Cancer – Ref#98) – were these the same sets of genes listed in Table S2 / column A?

There was an inconsistency between Table 2 and Table S2. These tables were corrected.

 Line 430 … please, add reference number (Ref#113?) after “Jiang et al.”

According to the reviewer’s comment, reference 113 was moved to after “Jiang et al”.

 Line 437 … please, add reference number (Ref#42?) after “rat liver.”

According to the reviewer’s comment, reference 42 was moved to after “rat liver”.

 Line 491 … define “HTP” – high throughput?

According to the reviewer’s comment, “HTP” was spelt out here, at first mention.

 Line 492 … “Buckalew et al.,” - missing dot

According to the reviewer’s comment, dot and comma were added after et al.

Table S1 … Heading of the Table is placed in the cell BL1, not A1

The heading of the Table S1 is now placed in A1.

 Table S2 … (Table S2: 60 genes differentially expressed in the breast cancer for selecting interacting chemicals) - this should be Table S3

There was an inconsistency of the supplementary Tables. We corrected the table numbers.

 Table S3 … (“cells/cell lines”) - this should be Table S2

There was an inconsistency of the supplementary Tables. We corrected the table numbers.

Reviewer 4 Report

The authors of the present manuscript reviewed the use of different transcriptomic platforms and approaches as pre-screening tools for the identification of relevant gene expression alterations related to non-genotoxic carcinogenesis. On the basis of these data it will be possible to select most suitable read-outs linked to specific Key events, thus supporting the weight of evidence, to be finally included in a IATA for non-genotoxic carcinogenesis. The manuscript is clear and well written and is suitable for publication in its current form except for very minor text corrections:

 - line 68: please make explicit “hprt” for those who are not familiar with this test

- line 195: I think you should change S2c in S1c as you changed the figure numbering

- line 258: please insert here the link to access the platform

- Figure 3: I think there is a typo: PI3K-AKT-mTOC should be mTOR

Author Response

Thank you for the positive comments. 

Our responses to the comments are below in red

- line 68: please make explicit “hprt” for those who are not familiar with this test

According to the reviewer's comment, we spelled out hprt and tk as well. 

- line 195: I think you should change S2c in S1c as you changed the figure numbering

Thank you for spotting this. We correct the figure number here. 

- line 258: please insert here the link to access the platform

According to the reviewer's comment, we inserted the link to the platform here.

- Figure 3: I think there is a typo: PI3K-AKT-mTOC should be mTOR

Thank you for spotting this. We have corrected the figure.